# Rational Generation of Monoclonal Antibodies Selective for Pathogenic Forms of Alpha-Synuclein

**DOI:** 10.3390/biomedicines10092168

**Published:** 2022-09-02

**Authors:** Ebrima Gibbs, Beibei Zhao, Andrei Roman, Steven S. Plotkin, Xubiao Peng, Shawn C. C. Hsueh, Adekunle Aina, Jing Wang, Clay Shyu, Calvin K. Yip, Sung-Eun Nam, Johanne M. Kaplan, Neil R. Cashman

**Affiliations:** 1Djavad Mowafaghian Centre for Brain Health, University of British Columbia, Vancouver, BC V6T 1Z1, Canada; 2Department of Physics and Astronomy, University of British Columbia, Vancouver, BC V6T 1Z1, Canada; 3Department of Biochemistry and Molecular Biology, University of British Columbia, Vancouver, BC V6T 1Z3, Canada; 4ProMIS Neurosciences, Cambridge, MA 02142, USA

**Keywords:** alpha-synuclein, oligomer, fibril, synucleinopathy, misfolding specific antibody, selectivity, conformational epitope, protein aggregation, sequence/structure determinants, aggregation-based technologies and therapeutics

## Abstract

Misfolded toxic forms of alpha-synuclein (α-Syn) have been implicated in the pathogenesis of synucleinopathies, including Parkinson’s disease (PD), dementia with Lewy bodies (DLB), and multiple system atrophy (MSA). The α-Syn oligomers and soluble fibrils have been shown to mediate neurotoxicity and cell-to-cell propagation of pathology. To generate antibodies capable of selectively targeting pathogenic forms of α-Syn, computational modeling was used to predict conformational epitopes likely to become exposed on oligomers and small soluble fibrils, but not on monomers or fully formed insoluble fibrils. Cyclic peptide scaffolds reproducing these conformational epitopes exhibited neurotoxicity and seeding activity, indicating their biological relevance. Immunization with the conformational epitopes gave rise to monoclonal antibodies (mAbs) with the desired binding profile showing selectivity for toxic α-Syn oligomers and soluble fibrils, with little or no reactivity with monomers, physiologic tetramers, or Lewy bodies. Recognition of naturally occurring soluble α-Syn aggregates in brain extracts from DLB and MSA patients was confirmed by surface plasmon resonance (SPR). In addition, the mAbs inhibited the seeding activity of sonicated pre-formed fibrils (PFFs) in a thioflavin-T fluorescence-based aggregation assay. In neuronal cultures, the mAbs protected primary rat neurons from toxic α-Syn oligomers, reduced the uptake of PFFs, and inhibited the induction of pathogenic phosphorylated aggregates of endogenous α-Syn. Protective antibodies selective for pathogenic species of α-Syn, as opposed to pan α-Syn reactivity, are expected to provide enhanced safety and therapeutic potency by preserving normal α-Syn function and minimizing the diversion of active antibody from the target by the more abundant non-toxic forms of α-Syn in the circulation and central nervous system.

## 1. Introduction

Strong genetic and experimental evidence supports a causative role for α-Syn in the pathogenesis of several progressive neurodegenerative disorders known collectively as synucleinopathies, including PD, DLB and MSA [1,2]. Current evidence indicates that α-Syn pathogenicity resides primarily with soluble, misfolded aggregates of the protein. In particular, oligomers and small soluble fibrils/protofibrils of α-Syn have been reported to mediate neurotoxicity and progression of disease [3,4], consistent with the prion-like propagation observed in other misfolding neurodegenerative diseases including Alzheimer’s disease and amyotrophic lateral sclerosis [5,6]. In contrast, α-Syn monomers and insoluble fibrils appear to carry little or no direct toxicity [3]. Lewy bodies and Lewy neurites containing insoluble fibrillar deposits of α-Syn are characteristic of disease but have been proposed to serve a protective role by sequestering toxic, misfolded aggregates of α-Syn away from the cellular machinery [4]. 

Oligomeric aggregates of misfolded α-Syn can exert toxicity via various mechanisms, including endoplasmic reticulum stress, membrane damage, disruption of mitochondrial and synaptic function, and promotion of inflammation [4]. Toxic oligomers also have the limited ability to propagate from cell to cell in a prion-like fashion as they are released into the extracellular space by neuronal cells either as free aggregates or carried by exosomes [7,8,9,10,11], and potentially in a direct cell-to-cell exchange via tunneling nanotubes [10]. However, small soluble fibrils/protofibrils of α-Syn are thought to be responsible for the bulk of seeding activity and can transmit α-Syn pathology in vitro and in vivo [12,13,14]. In vitro, the addition of sonicated preparations of PFFs to neuronal cells leads to the recruitment of endogenous α-Syn into misfolded, phosphorylated aggregates [14]. In vivo, injection of PFFs or aggregated α-Syn from diseased patient brains into the CNS of rodents leads to the progressive spread of intracellular pathogenic α-Syn aggregates from the site of injection to other connected areas of the brain [12,13]. 

Selective targeting of pathogenic forms of α-Syn with antibodies represents an attractive therapeutic strategy. The advantage of selective antibodies, as opposed to a pan α-Syn targeting approach, lies in preserving normal α-Syn function and minimizing the diversion of active antibody from the target by non-toxic forms of the protein. Systemic delivery of selective antibodies has the potential to inhibit cell-to-cell propagation of toxic aggregates in the extracellular space [14,15,16] as well as causing intracellular degradation of pathogenic α-Syn aggregates after internalization [17,18]. In this context, selective antibodies that avoid binding to abundant α-Syn monomers in the blood, brain interstitial fluid, and cerebrospinal fluid (CSF) have the potential to achieve greater therapeutic effectiveness and reduce the risk of infusion reactions. 

Intracellular delivery of intrabody constructs using gene therapy vectors offers the possibility of clearing pre-existing cytoplasmic aggregates and inhibiting further propagation at the source. Stringent selectivity for pathogenic α-Syn species is of particular importance in this context to avoid interference with intracellular α-Syn monomers, which play an important role in the regulation of synaptic vesicle trafficking and neuronal survival [19], and with physiologic tetramers involved in maintaining α-Syn homeostasis and inhibiting the formation of pathogenic aggregates [20,21,22].

In order to generate optimally selective antibodies against pathogenic forms of α-Syn, we used computational modeling to identify conformational epitopes predicted to be exposed in α-Syn oligomers and small soluble fibrils/protofibrils, but to be inaccessible in large fully formed insoluble fibrils or α-Syn monomers [23]. Cyclic peptide scaffolds reproducing the conformational epitopes were used to immunize mice and generate mAbs. Our results indicate that this approach allows for the efficient generation of antibodies with the desired selective binding profile and protective activity against toxic and propagating pathogenic α-Syn species. 

## 2. Materials and Methods

### 2.1. Generation of Monoclonal Antibodies 

Computational modeling using Collective Coordinates [24] predicted exposure of conformational epitopes in the EKTKEQ (aa 57–62) region of misfolded pathogenic species of α-Syn (oligomers, small soluble fibrils) but not on α-Syn monomers or fully formed insoluble fibrils [23]. The conformational ensembles of the candidate epitopes EKTK and TKEQ predicted by the algorithm were approximated with cyclic peptide scaffolds containing flanking glycine residues on each side (“glycindels”) [25], and a cysteine between the glycine linkers, for conjugation to a carrier protein for immunization. Cyclization was performed via a head-to-tail (C–G) amide bond to generate representative constructs predicted to have the desired conformational profile: cyclo(CGGGEKTKGG), cyclo(CGGGGEKTKGG), cyclo(CGTKEQGGGG) and cyclo(CGGTKEQGGGG). The constructs were then conjugated to keyhole limpet hemocyanin (KLH) or bovine serum albumin (BSA) via maleimide-based coupling. Non-cyclized, linear peptides with the same sequence were also synthesized as conformational negative control peptides. Peptide synthesis was performed by CPC Scientific Inc. (Sunnyvale CA, USA) following standard manufacturing procedures.

Immunization of mice for generation of mAbs, initial screening of hybridoma clones by enzyme-linked immunosorbent assay (ELISA) and antibody purification were performed by ImmunoPrecise Antibodies (Victoria, BC, Canada). Balb/c mice were immunized with cyclic peptide conjugated to KLH and splenic lymphocytes were collected 19 days later for hybridoma cell line generation. Tissue culture supernatants from the hybridomas were screened by ELISA for reactivity to cyclic peptide conjugated to BSA and lack of reactivity to BSA-conjugated linear peptide, BSA alone, or human transferrin as an irrelevant antigen. Briefly, plates were coated overnight with 100 µL/well of 0.1 µg/mL BSA-conjugated cyclic peptide, BSA-conjugated linear peptide or BSA alone in carbonate buffer (pH 9.6) at 4 °C, or with 50 µL/well of 0.25 µg human transferrin in distilled water at 37 °C. Hybridoma supernatant reactivity (100 µL/well) was detected by the addition of horseradish peroxidase (HRP)-conjugated goat anti-mouse IgG secondary antibody followed by the substrate 3,3′,5,5′-tetramethylbezidine (TMB). Absorbance was read at 450 nM. Selected clones with the desired reactivity were expanded in culture and immunoglobulin was purified using Protein G. All mAbs were resuspended in low endotoxin phosphate-buffered saline (PBS) for further testing.

### 2.2. Preparations of Alpha-Synuclein 

Recombinant full length monomeric human α-Syn peptide was obtained from rPeptide (Athens, GA, USA). Toxic oligomers were prepared by incubating α-Syn monomers (4 µM) at 37 °C with slow agitation (approximately 8 rpm) for 3 days. These oligomer preparations consisted almost exclusively of tetramers as determined by size exclusion chromatography (SEC, Appendix A). Toxicity of the oligomers was assessed as described below. Physiologic tetramers were prepared by incubating monomeric α-Syn peptide (5 mg/mL) with shaking at 1000 rpm at 37 °C for 7 days and collecting aggregation-resistant α-Syn in the supernatant. The preparation was confirmed to contain primarily tetramers by SEC (Appendix A). Full length human α-Syn PFFs were purchased from StressMarq (Victoria, BC, Canada, SPR-322) and were sonicated to generate small soluble seeding fibrils as described previously [12,13,26]. Immediately before use, PFFs were sonicated in a Branson Ultrasonic water bath for 5 min followed by gently pipetting up and down. 

### 2.3. Brain Extract Preparation 

Post-mortem brain tissues from DLB and MSA patients were obtained from brain banks affiliated with the University of Calgary (kind gift of Dr. Jeffrey Joseph) and the University of British Columbia (kind gift of Dr. Matthew Farrer), respectively. Informed consent for tissue collection at autopsy and neurodegenerative research use was obtained in accordance with local institutional review boards. These studies were reviewed and approved by the UBC Ethics Board and are in accordance with the Declaration of Helsinki principles. Samples from the cerebellum of MSA patients and frontal cortex anterior cingulate of DLB patients were submersed in ice-cold Tris-buffered saline (TBS) (20% *w/v*) with EDTA-free protease inhibitor cocktail (Roche Diagnostics, Laval, QC, Canada), and homogenized using an Omni tissue homogenizer (Omni International Inc, Keenesaw, GA, USA), 3 × 30 s pulses with 30 s pauses in between, all performed on ice. Homogenates were then subjected to ultracentrifugation at 100,000× *g* for 60 min. Supernatants (soluble extracts) were collected, aliquoted and stored at −80 °C. The protein concentration was determined using a bicinchoninic acid (BCA) protein assay.

### 2.4. Antibody Binding Assays 

Binding of mAbs to a range of concentrations of α-Syn monomers, toxic oligomers and sonicated PFFs was measured at Millipore Sigma (Hayward, CA, USA) using the SMCxPro^TM^ immunoassay. Briefly, magnetic particles were coated with the test mAbs acting as capture antibodies, and the captured analyte was detected with a fluorescently labeled pan α-Syn-reactive antibody as the secondary detector antibody (clone 4D6, BioLegend, Dedham, MA, USA, previously Covance Cat.# SIG-39720).

SPR analysis was performed on a MASS-2 instrument (Bruker Daltronics, Billerica MA, USA) to measure mAb binding to various synthetic α-Syn species and to endogenous α-Syn in brain extract from DLB and MSA patients. Test antibodies were immobilized on high amine capacity sensorchips at a density of approximately 9000–11,000 response units (RUs), and α-Syn monomers, toxic oligomers, sonicated PFFs and physiologic tetramers were injected over the surfaces at a concentration of 10 µM for 8 min at 10 µL/min. The molarity of α-Syn in these preparations was calculated based on the molecular weight of the monomer. For measurement of binding to brain extracts, test antibodies were immobilized on sensorchips at a density of 11,000–20,000 RUs and brain extracts diluted to 200 µg/mL were injected over the surfaces for 8–10 min at 10 µL/min. The binding responses from the resultant sensorgrams were double-referenced against unmodified reference surfaces and blank buffer injections. SPR results are expressed as binding response units (BRUs) obtained post-injection at 30 s into the dissociation phase to provide a quantitative assessment of overall antibody–ligand interactions in samples with multiple, heterogeneous species present at undefined concentrations such as in sonicated PFFs and soluble brain extracts. 

### 2.5. Immunohistochemistry 

Fresh frozen brain sections with no fixation were exposed to antigen retrieval citrate buffer (Target Retrieval Solution, Dako, Santa Clara, CA, USA) for 20 min and incubated in a humidified chamber with serum-free protein blocking reagent (Dako) for 1 h to block non-specific staining. The sections were incubated with test mAbs, a pan α-Syn-reactive antibody (clone 4D6, BioLegend), or mIgG1 isotype control at 20 µg/mL overnight at 4 °C. Sections were then washed three times in TBS with 0.1% Triton-X-100 (TBS-T) followed by incubation with HRP-conjugated sheep anti-mouse IgG secondary antibody (Cytiva, Vancouver, BC, Canada, NA931) for 1 h at room temperature and three washes in TBS-T. Secondary antibody was also added to sections that were not exposed to primary antibody as a negative control. The HRP enzyme substrate, biaminobezidine (DAB) chromogen reagent was then added to the sections, followed by rinsing with distilled water. The sections were counterstained with haematoxylin QS (Vector Laboratories, Burlingame, CA, USA). The slides were examined under a light microscope (Zeiss Axiovert 200 M, Carl Zeiss Toronto, ON, Canada) and representative images were captured using a Leica DC300 digital camera and software (Leica Microsystems Canada Inc., Vaughan, ON, Canada). 

### 2.6. Thioflavin-T Seeding Assay 

Monomeric human α-Syn peptide at 100 µM was incubated in PBS (pH 7.4) with 25 µM thioflavin-T (ThT) in the presence or absence of various seeding agents including BSA-conjugated cyclized and linear peptide epitopes (100 nM), sonicated PFFs (10 nM), oligomers (100 nM) and physiologic tetramers (100 nM) in a 120 µL reaction volume per well of black-walled 96-well microtiter plates (Greiner Bio-One, Monroe, NC, USA). In studies of neutralizing activity, mAb was added at 0.1 nM. Plates were incubated at 37 °C with shaking for 30 sec prior to each hourly reading of ThT fluorescence (excitation at 440 nm, emission at 486 nm) using a Wallac Victor3v 1420 Multilabel Counter (PerkinElmer, Waltham, MA, USA). To be noted, in some studies, continuous shaking was used to promote aggregation, giving rise to higher levels of aggregation with monomers alone. 

### 2.7. Electron Microscopy of Fibrils

Fibrils were adsorbed to glow discharged carbon-coated copper grids and stained with uranyl formate. The negative stain specimens were examined on a Talos L120C transmission electron microscope (Thermo Fisher Scientific, Waltham, MA, USA) operating at an accelerating voltage of 120 kV and equipped with a 4 K Ceta CMOS camera. Micrographs were acquired at a nominal magnification of 45,000× *g* at a defocus of approximately 1 µm, and the widths of the observed fibrils were measured using ImageJ. Measurements were taken for 23, 44, and 30 individual fibrils for PFFs, cyclic peptides 2.2 and 3, respectively. 

### 2.8. In Vitro Neurotoxicity Assay 

In vitro assessment of neurotoxicity was performed by Neuron Experts (Marseille, France) following established protocols. Rat dopaminergic neurons were derived from the midbrains of embryonic day 15 Wistar rat fetuses and cultured at 40,000 cells/well in 96-well plates pre-coated with poly-d-lysine (Thermo Fisher, Lille, France). In toxicity assays, α-Syn oligomers (0.5 µM) prepared as described above, or BSA-conjugated conformational cyclized peptide or the corresponding linear peptide (0.1, 1.0, 5.0 µM) were added to the cultures. In studies of neurotoxicity inhibition by mAbs, α-Syn oligomers were pre-incubated for 30 min with or without test mAbs (0.25 µM) and added to neuronal cultures. Control wells with mAbs alone were included to test for any potential toxicity of the mAbs themselves. Brain-derived neurotrophic factor (BDNF, 50 ng/mL) was used as a positive control. All conditions were tested in 6 replicate wells. Cultures were incubated for 4 days at 37 °C in humidified air and 5% CO_2_ and the cells were fixed with 4% paraformaldehyde followed by permeabilization and blocking of non-specific binding sites with a solution of PBS containing 0.1% saponin and 1% fetal calf serum. The cells were then incubated with monoclonal chicken anti-tyrosine hydroxylase (Abcam, Cambridge, MA, USA, ab76442) followed by Alexa Fluor 488-conjugated goat anti-chicken IgG (Thermo Fisher, Lille, France, 13417227) to stain dopaminergic neurons. Cell nuclei were stained with fluorescent Hoechst dye. Twenty pictures per well were taken using an InCell AnalyzerTM 2200 (GE Healthcare, Chicago, IL, USA) under 20× magnification for quantitation of TH positive neurons using Developer software (GE Healthcare, Chicago, IL, USA). 

### 2.9. Cellular Seeding Assay

In vitro assessment of the ability of mAbs to inhibit the uptake and promotion of intracellular α-Syn aggregate formation by human PFFs was performed by Neuron Experts (Marseille, France) following established protocols. Rat hippocampal neurons were derived from the hippocampi of embryonic day 17 Wistar rat fetuses and cultured at 20,000 cells/well in 96-well plates pre-coated with poly-d-lysine (Thermo Fisher, Lille, France). Sonicated PFFs (1 µg/mL), prepared as described above, were pre-incubated for 30 min with or without mAbs (0.05 µM) and added to neuronal cultures. All conditions were tested in 6 replicate wells. Cultures were incubated for 14 days at 37 °C in humidified air and 5% CO_2_. Medium was changed once a week without new addition of PFFs or mAbs. At the end of the incubation period, cells were fixed in 4% paraformaldehyde/4% sucrose/1% Triton X-100 followed by permeabilization and blocking of non-specific binding sites with a solution of PBS containing 0.1% Triton X-100 and 3% BSA. For detection of neurons with aggregates of human α-Syn resulting from human PFF uptake, the cells were incubated with primary chicken anti-microtubule-associated protein 2 (MAP2, Abcam, Cambridge, MA, USA, ab5392) and rabbit anti-human α-Syn (Thermo Fisher, Lille, France, 701085). For detection of neurons with endogenous, phosphorylated aggregates of rat α-syn, the cells were incubated with primary chicken anti-MAP2 (Abcam, Cambridge, MA, USA) and rabbit anti-phosphorylated Ser129 rat α-Syn (Abcam, ab51253). Bound antibodies were visualized by staining with secondary Alexa Fluor 568-conjugated goat anti-chicken IgG (Thermo Fisher, Lille, France, A-11041) and Alexa Fluor 633-conjugated goat anti-rabbit IgG (Thermo Fisher, A-21070). Cell nuclei were stained with fluorescent Hoechst dye. Twenty pictures per well were taken using an InCell AnalyzerTM 2200 (GE Healthcare) under 20× magnification for quantitation of MAP2 positive neurons with α-Syn aggregates using Developer software (GE Healthcare). 

### 2.10. Statistics

Statistical analysis was conducted with GraphPad Prism (GraphPad Software, San Diego, CA, USA). A global analysis of the data was performed using one-way analysis of variance (ANOVA) followed by Dunnett’s multiple comparisons test. 

## 3. Results

### 3.1. Identification of Pathogenic Alpha-Synuclein Conformational Epitopes

The Collective Coordinate (CC) computational algorithm [24] was applied to stressed α-Syn protofibrils (PDB ID: 2N0A) to identify conformational epitopes predicted to be solvent-exposed on pathogenic forms of α-Syn but not on monomers or on fully formed fibrils. The CC algorithm identified EKTKEQ (aa 57–62) as a region thermodynamically likely to be solvent-exposed in oligomers and small soluble fibrils, with less inter-chain electrostatic and van der Waals interactions with neighboring monomers in the fibril, and increased local dynamics [23]. Cyclic peptide scaffolds containing a variable number of glycines at N- and C-termini (“glycindels”) were constructed to mimic the conformation of the predicted EKTK and TKEQ epitopes present in the oligomer (Figure 1). These glycindels were then conjugated to a carrier protein for immunization.

To assess the biological relevance of the predicted epitopes, a representative peptide scaffold, cyclo (CGTKEQGGGG), was tested for its ability to replicate the seeding activity and neurotoxicity of small soluble fibrils and α-Syn oligomers, respectively (Figure 2). Seeding activity was tested in a Thioflavin-T (ThT) fluorescence assay, which measures the fibrillogenic aggregation of α-Syn protein monomers over time in the absence or presence of peptide. While monomers incubated alone did not undergo detectable aggregation under the conditions tested, the presence of the cyclic peptide scaffold promoted aggregation, as shown by a steady increase in fluorescence signal after an initial lag period, reaching a plateau approximately 120 h later. In contrast, the corresponding linear peptide did not exhibit any seeding activity (Figure 2a). In addition, negative stain electron microscopy analysis of the material recovered in the plateau phase at the end of the incubation period indicated that the morphology of fibrils seeded by peptide scaffolds resembles that of fibrils seeded with sonicated PFFs (Appendix A). Notably, the average widths of fibrils seeded by peptide scaffolds (12.2 ± 1.5 nm for cyclo (CGGKTKEGG) and 11.3 ± 1.6 nm for cyclo (CGGEKTKGGG) are consistent with those of fibrils seeded by PFFs (11.5 ± 1.6 nm).

In cultures of rat primary dopaminergic neurons, the conformational cyclic peptide scaffold by itself exhibited toxicity comparable to that of α-Syn oligomers. In contrast, the corresponding linear version of the same peptide was devoid of toxicity and did not affect neuronal viability (Figure 2b).

Taken together, these results suggest that small misfolded regions (our predicted conformational epitopes) exposed on toxic α-Syn aggregates may directly contribute to their pathogenicity and represent biologically relevant targets for therapeutic antibodies.

### 3.2. Selectivity of Monoclonal Antibodies Raised against Conformational Epitopes 

Characterization of mAbs generated by immunization with predicted conformational peptide epitopes showed selective binding to pathogenic but not physiologic forms of α-Syn. Measurement of antibody binding using the Millipore SMC platform showed robust interaction of mAbs with preparations of α-Syn toxic oligomers and small soluble fibrils in the pg/mL range. By comparison, there was little or no binding of the mAbs to α-Syn monomers up to a µg/mL range, representing a 9000–175,000-fold difference in selectivity for pathogenic species vs. monomers (Figure 3).

A selective binding profile was also observed using SPR analysis. Immobilized mAbs all showed similar robust binding to α-Syn toxic oligomers and small soluble fibrils with little or no binding to monomers or physiologic tetramers (Figure 4). By comparison, a commercial pan α-Syn antibody showed comparable binding to all forms of α-Syn.

Significantly, the preparations of toxic oligomers used in the assays were found to consist primarily of tetramers as determined by SEC (Appendix A) and displayed seeding activity (Figure 5a). In contrast, physiologic tetramer preparations behaved in a manner consistent with that described for physiologic tetramers derived from cell lysates and did not show seeding activity (Figure 5a), in line with their reported role in the maintenance of α-Syn homeostasis and resistance to aggregation [22]. SPR analysis of binding responses indicated that mAbs raised against the conformational epitopes have the striking ability to distinguish between toxic tetramers and physiologic tetramers (Figure 4 and Figure 5b). This is in agreement with reports on the structure of tetramers, suggesting that the EKTK and TKEQ target epitopes are expected to be buried within the hydrophobic core of properly folded physiologic tetramers, making them unavailable for binding by the mAbs but presumably becoming exposed and accessible to the mAbs in misfolded toxic tetramers [27,28,29]. 

The reactivity of mAbs with Lewy bodies containing insoluble fibril deposits of α-Syn was evaluated by immunohistochemical staining of brain sections from patients with dementia with Lewy bodies (DLB). As expected, Lewy bodies were recognized and stained by a control pan α-Syn antibody. By comparison, mAbs showed occasional staining of small aggregates but no appreciable reactivity with Lewy bodies, in line with the computationally predicted lack of target epitopes on fully formed, insoluble fibrils (Figure 6).

Finally, the reactivity of mAbs with native pathogenic α-Syn species in patient brain extract was assessed by SPR analysis. As shown in Figure 7, immobilized mAbs showed robust binding to α-Syn in soluble brain extract from patients diagnosed with DLB (Figure 7a) or MSA (Figure 7b). With MSA brain extract, the magnitude of binding by all mAbs was equivalent to or greater than that of a control pan α-Syn antibody capable of binding both normal and pathogenic α-Syn species.

Overall, these results indicate that mAbs raised against predicted conformational epitopes of pathogenic α-Syn possess substantial selectivity. The mAbs show binding to preparations of toxic oligomers and small soluble fibrils while sparing α-Syn monomers and physiologic tetramers important for neuronal function. The mAbs also show little interaction with insoluble fibrillar brain deposits of α-Syn, which could otherwise act as a sink diverting antibodies away from their pathogenic target. 

### 3.3. Protective Activity of Monoclonal Antibodies Raised against Conformational Epitopes 

In vitro assays were conducted to determine whether binding of mAbs to α-Syn oligomers and small soluble fibrils observed above translated into inhibition of their pathogenic activity. 

First, a ThT fluorescence assay was performed to examine the ability of mAbs to inhibit the seeding activity of preformed small soluble fibrils (sonicated PFFs). As shown in Figure 8, human PFFs acted as a potent seeding agent promoting the aggregation of α-Syn monomers over time, as expected. The addition of mAb completely abrogated the seeding activity of PFFs (1:100 molar ratio of mAb:PFF). 

Second, rat primary dopaminergic neurons were exposed to α-Syn oligomers without or with mAbs. As shown in Figure 9, α-Syn oligomers resulted in a loss of viability (number of viable neurons 63% of vehicle control). The addition of mAbs to the cultures provided significant protection against neuronal toxicity, comparable to that obtained with the positive control, brain-derived neurotrophic factor (BDNF) (Figure 9). In cultures of rat primary hippocampal neurons, mAbs significantly reduced the uptake of human PFFs and subsequent formation of intracellular human α-Syn aggregates (Figure 10). Moreover, in the same cultures, the mAbs also reduced the recruitment of endogenous rat α-Syn into pathogenic phosphorylated aggregates (Figure 11). 

Together, the observed inhibition of fibril seeding activity and oligomer toxicity by the mAbs agrees with the fact that the conformational epitopes targeted, on their own, possessed seeding activity and toxicity. 

## 4. Discussion

In this study, computational modeling was used to predict conformational epitopes exposed on pathogenic forms of α-Syn (oligomers and small soluble fibrils) but not on physiologically important forms of the protein (monomers, physiologic tetramers) or on largely non-toxic insoluble fibrillar deposits of α-Syn (Lewy bodies, Lewy neurites). Remarkably, small peptide scaffolds computationally predicted to reproduce the conformational epitopes exhibited neurotoxicity and seeding activities like those of the fully formed protein species (Figure 2), suggesting that these misfolded regions play a direct role in the pathogenicity of α-Syn and represent both immunologically and biologically relevant targets. The mAbs generated by immunization with the conformational peptide scaffolds showed a high degree of selectivity, capable of distinguishing not only between monomers and soluble oligomers and fibrils of α-Syn (Figure 3 and Figure 4), but also between aggregation-resistant physiologic tetramers and toxic tetramers (Figure 4 and Figure 5b). The ability to distinguish between these closely related species is likely due to the conformational nature of the EKTK and TKEQ epitopes, which presumably become exposed on misfolded tetramers but are hidden within the core of properly folded tetramers as suggested by structural studies [27,28,29]. As also predicted by our computational design, the mAbs showed undetectable reactivity with Lewy bodies in DLB brain sections (Figure 6), unlike most other α-Syn antibodies described in the literature [15,16,30,31,32,33]. 

In line with our evidence that the target epitopes possess pathogenic activities, the mAbs raised inhibited the neurotoxicity of α-Syn oligomers in cultures of primary rat dopaminergic neurons (Figure 9) and inhibited the seeding activity of small soluble fibrils of α-Syn (sonicated PFFs) in both a cell-free ThT assay (Figure 8) and cultures of primary rat hippocampal neurons (Figure 10 and Figure 11). These results indicate that computational modeling followed by immunization with disease-associated epitopes represents an efficient strategy to generate selective and protective mAbs against pathogenic α-Syn. 

Traditional methods of immunization using linear peptide epitopes, whole protein or protein aggregates, typically give rise to antibodies that recognize all forms of the protein. Although some α-Syn-directed mAbs have been described to show greater selectivity for oligomers, significant binding to monomers is still observed along with prominent reactivity with Lewy bodies [15,16,30,31,32,33]. In comparison, the mAbs with selectivity for toxic α-Syn species described in the present study present an advantage in that they can both avoid interference with normal α-Syn function and prevent off-target binding to non-toxic forms of α-Syn, which could otherwise act as a sink to reduce the effective dose. Based on the current understanding of α-Syn biology, selective antibodies possess a strong potential to provide greater efficacy and safety in both the context of systemic administration and intracellular delivery as an intrabody genetic construct. 

Although toxic aggregates of α-Syn form intracellularly, preclinical studies in transgenic mouse models of PD [14,15,16,30,32], DLB [18] and MSA [34] have shown that passive administration of α-Syn-directed antibodies can promote clearance of pathogenic α-Syn, ameliorate neurodegeneration and provide a functional benefit. Mechanistic studies indicate that antibodies administered systemically can act by neutralizing and clearing α-Syn aggregates as they propagate in the extracellular space [14,15,16], and via internalization of antibodies by neurons allowing for binding and degradation of intracellular α-Syn aggregates through autophagy or lysosomal digestion [17,18]. When delivered extracellularly, antibodies selective for pathogenic α-Syn species present the advantage of minimizing binding to monomers present in the circulation at ng-µg/mL levels, thereby maintaining the effective dose and reducing the risk of infusion reactions. Once across the blood-brain barrier, avoidance of binding to α-Syn monomers in the interstitial space (pg-ng/mL) or insoluble fibrils released from dying neurons would similarly act to focus the dose of antibodies on propagating or membrane-associated pathogenic aggregates. To date, Phase 2 trials with intravenous delivery of the α-Syn-directed mAbs, prasinezumab and cinpanemab in PD patients have failed to show significant efficacy against the primary endpoint of MDS-UPDRS total score, a measure of impairment and disability [35]. The observed lack of efficacy may have been due in part to suboptimal selectivity of the antibodies for pathogenic α-Syn species or the choice of target epitope. Further development of cinpanemab has been abandoned but testing of prasinezumab continues based on a positive signal with secondary and exploratory endpoints. 

In the context of intracellular delivery of intrabody via gene therapy vectors, selectivity for pathogenic α-Syn species within the cell is not only an advantage but also a requirement to maintain normal α-Syn function critical to neuron survival. The α-Syn is an abundant protein with a predominantly pre-synaptic location where it interacts with membranes and is believed to play a key role in neurotransmitter release, dopamine metabolism, maintenance of synaptic pool vesicles and vesicle trafficking, as well as overall synaptic activity and plasticity [19]. Recent evidence indicates that α-Syn exists in equilibrium between cytosolic unfolded monomers and aggregation-resistant physiologic helical tetramers, and that disruption of tetramer formation results in the presence of excess monomers which promote the formation of misfolded toxic oligomers [20,21,22]. Expression of α-Syn with tetramer-abrogating mutations in the KTKEGV motif, was found to increase the concentration of aggregation-prone monomers leading to neurotoxicity in vitro [21] and causing PD-like symptoms and pathologies in transgenic mice [20]. These observations suggest that antibodies such as the mAbs generated here, which can distinguish between physiologic tetramers and toxic tetramers/oligomers, will be required for the safe application of intrabodies. 

While the role of propagating toxic aggregates of α-Syn in the pathogenesis of synucleinopathies is now well-recognized, there is evidence that such aggregates may exist as different strains with different physical properties and seeding activity. For example, it has been reported that different strains of α-Syn fibrils can be isolated from the brains of PD and DLB patients (inclusions in neurons) compared to MSA patients (inclusions primarily in oligodendrocytes) [36,37,38]. 

Our early results showing reactivity of the same antibodies with extracts from both DLB and MSA brains (Figure 7) suggest that conformational epitopes on misfolded, toxic α-Syn may be shared across strains. This is like our previous observations with amyloid-beta showing reactivity of the oligomer-specific PMN310 antibody with extracts from multiple Alzheimer’s disease brains [39]. 

Overall, our results demonstrate that immunization with conformational epitopes of toxic α-Syn predicted by computational modeling allowed for the successful generation of highly selective mAbs capable of inhibiting the seeding activity and neurotoxicity of pathogenic α-Syn while avoiding detrimental interaction with normal monomers and physiologic tetramers. The use of selective antibodies delivered systemically or as intrabodies via gene therapy vectors is expected to provide an improved efficacy and safety profile compared to a pan α-Syn approach. 

## Figures and Tables

**Figure 1 biomedicines-10-02168-f001:**
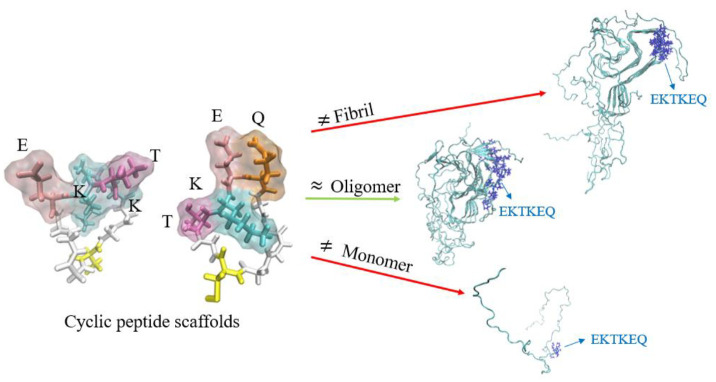
Alpha-synuclein residues 57–60 (EKTK) and 59–62 (TKEQ), held in a constrained turn, represent conformational epitopes predicted to be exposed on oligomers and small soluble fibrils but not monomers or fully formed insoluble fibrils. Representative molecular dynamics simulated conformations of the cyclic epitopes with the side chains oriented into solvent are shown.

**Figure 2 biomedicines-10-02168-f002:**
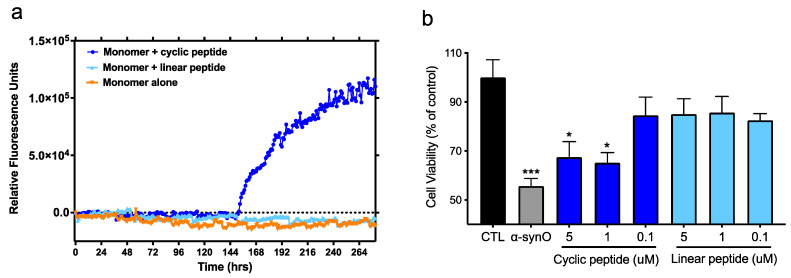
Conformational cyclic peptide epitopes have seeding activity and toxicity. (**a**) Aggregation was tracked hourly (shaking for 30 s prior to each reading) with a ThT fluorescent assay after the addition of α-syn monomers alone (100 mM) or in the presence of BSA-conjugated cyclized or linear CGTKEQGGGG peptide (100 nM). Data are representative of 3 independent experiments. (**b**) Viability of primary rat dopaminergic neurons treated with α-syn oligomers (0.5 μM) or increasing concentrations of BSA-conjugated cyclized or linear peptide. CTL = neurons incubated with vehicle alone. Mean ± SEM of 6 replicates. Global analysis of data performed using one-way ANOVA followed by Dunnett’s multiple comparisons test. * *p* ≤ 0.05, *** *p* ≤ 0.001 vs. CTL (vehicle). No statistically significant difference between linear peptide and CTL or between α-syn oligomers and 1, 5 μM cyclic peptide.

**Figure 3 biomedicines-10-02168-f003:**
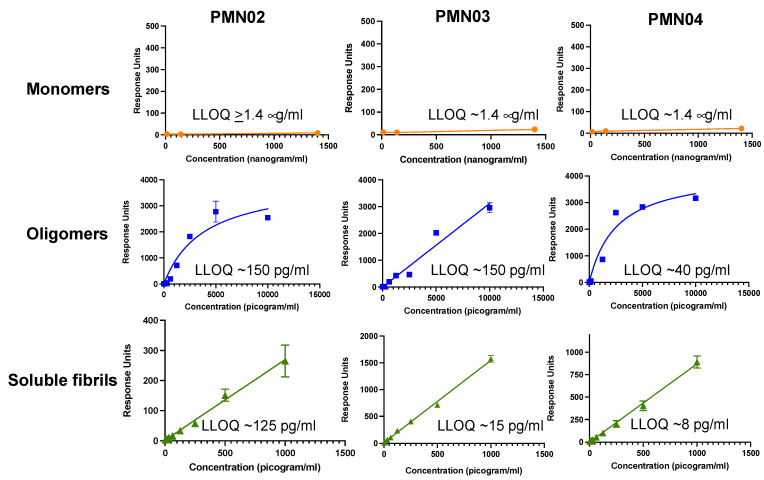
Selectivity of mAbs for pathogenic species of α-syn. The binding response of mAbs to various concentrations of α-syn monomers, toxic oligomers and soluble fibrils (sonicated PFFs) was measured in a Millipore immunoassay. Mean ± SD of triplicates shown with the calculated lower limit of quantitation (LLOQ) for each species.

**Figure 4 biomedicines-10-02168-f004:**
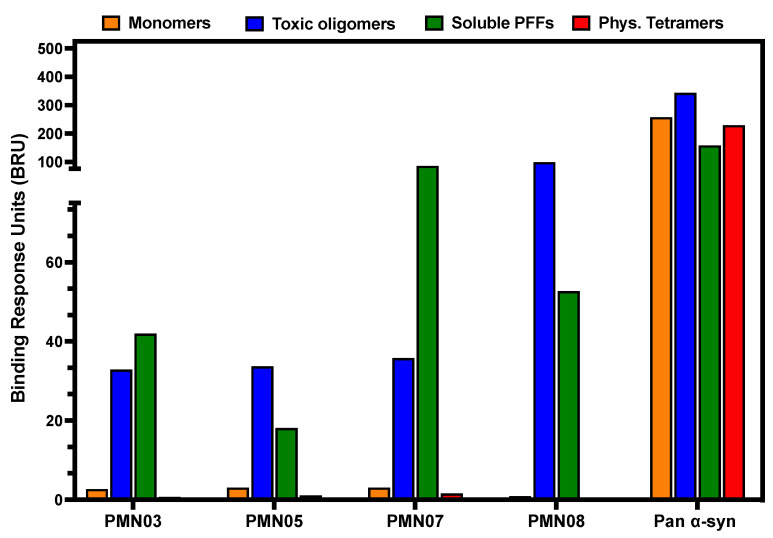
Selective binding of mAbs to pathogenic species of α-syn by SPR. The binding response of immobilized mAbs to α-syn monomers, toxic oligomers, soluble (sonicated) PFFs and physiologic (Phys.) tetramers was measured by SPR. An antibody reactive with all species of α-syn (Pan α-syn) was used as a positive control. The same pattern of binding was observed in 4 independent experiments.

**Figure 5 biomedicines-10-02168-f005:**
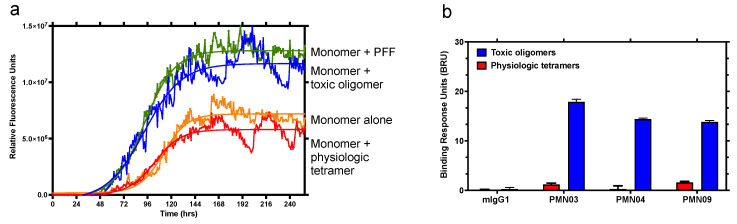
Ability of mAbs to distinguish between toxic oligomers and physiologic tetramers. (**a**) Aggregation was tracked hourly with a ThT fluorescent assay after the addition of α-syn monomers alone (100 mM) or in the presence of PFF (10 nM), toxic oligomers or physiologic tetramers (both 100 nM). Continuous shaking was used to promote aggregation, giving rise to measurable aggregation with monomers alone. Data are representative of 2 independent experiments. (**b**) The binding response of immobilized mAbs and negative control mouse IgG1 (mIgG1) to toxic oligomers and physiologic tetramers was measured by SPR. The same pattern of binding was observed in 4 independent experiments. Error bars: SEM.

**Figure 6 biomedicines-10-02168-f006:**
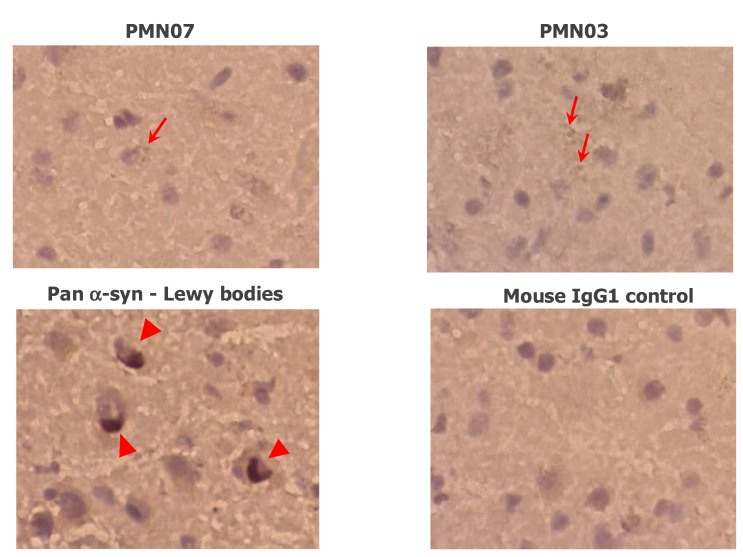
Reactivity of mAbs with DLB brain sections. mAbs show occasional staining of small aggregates (*arrows*) as opposed to the dense insoluble deposits of α-syn in Lewy bodies stained by a pan α-syn reactive antibody (*arrowheads*). No staining is seen with a mouse IgG1 negative control. Images are representative of the staining pattern seen with 17 mAbs tested. Magnification: 40×.

**Figure 7 biomedicines-10-02168-f007:**
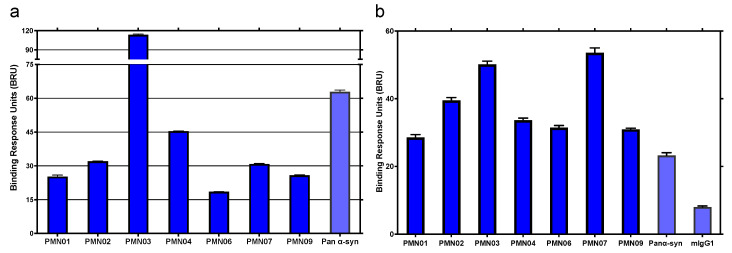
Binding of mAbs to native pathogenic α-syn species in patient brain extract. The binding response of immobilized mAbs to α-syn in brain extractS from DLB (**a**) and MSA (**b**) patients was measured by SPR. A pan α-syn reactive antibody and mouse IgG1 (mIgG1) were used as controls. Results shown are the mean ± SEM of 2 (**a**) or 4 (**b**) independent studies.

**Figure 8 biomedicines-10-02168-f008:**
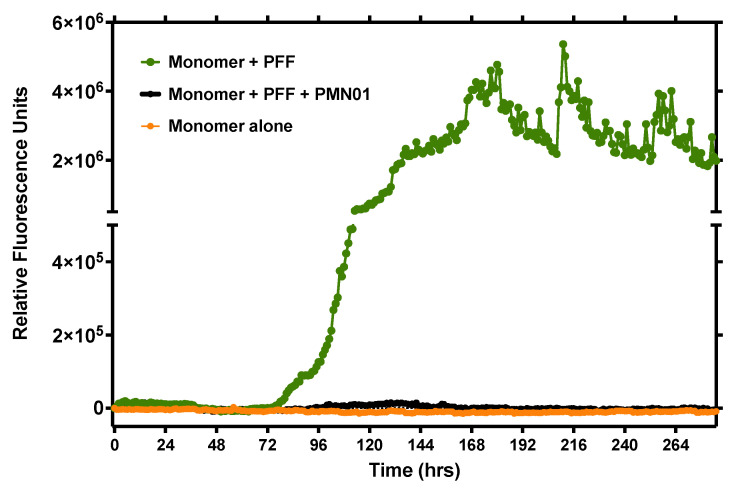
mAb inhibition of PFF seeding activity. Aggregation was tracked hourly with a ThT fluorescent assay (shaking for 30 sec prior to each reading) after the addition of α-syn monomers alone (100 mM) or in the presence of sonicated human PFF (10 nM) as a seeding agent, without or with mAb (0.1 nM). Data are representative of 2 independent experiments.

**Figure 9 biomedicines-10-02168-f009:**
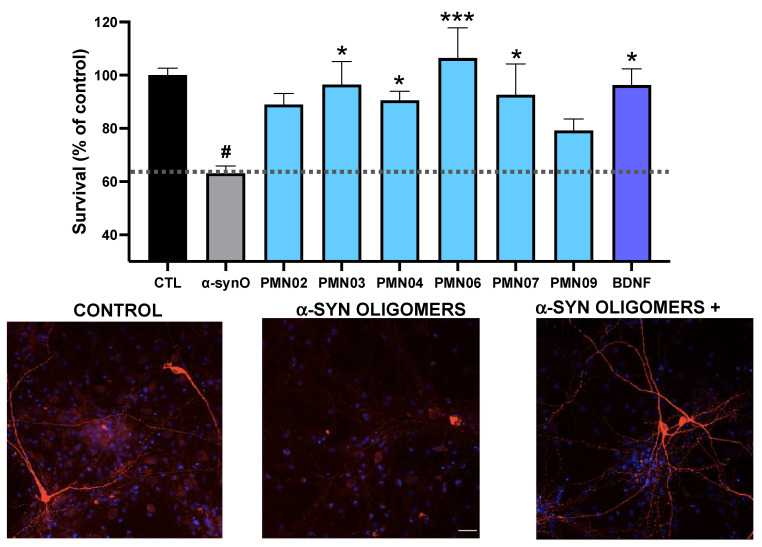
mAb inhibition of oligomer toxicity in dopaminergic neurons. Cultures of primary rat dopaminergic neurons were exposed to toxic α-syn oligomers (α-synO; 0.5 μM) without or with mAbs (0.25 μM). Survival is expressed as the percentage of viable neurons compared to a control culture with vehicle only (CTL). BDNF was used as a positive control. Results shown are the mean ± SEM of 6 replicate cultures. A global analysis of the data was performed using one-way ANOVA followed by Dunnett’s multiple comparisons test. # *p* < 0.005 vs. CTL, * *p* ≤ 0.05 vs. α-synO, *** *p* ≤ 0.001 vs. α-synO. No statistically significant difference between BDNF and CTL or mAbs and CTL. Representative images of cultures with staining for neurons (MAP2, *red*) and nuclei (*blue*) are shown.

**Figure 10 biomedicines-10-02168-f010:**
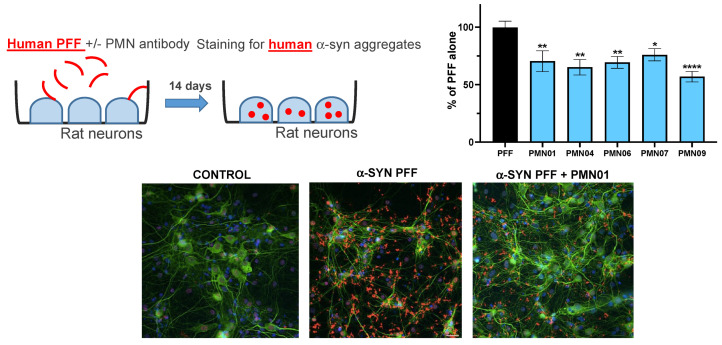
mAb inhibition of PFF uptake and intracellular aggregate formation. Cultures of primary rat hippocampal neurons were exposed to sonicated human PFF (1 μg/mL) without or with mAbs (0.05 μM, except for PMN09 at 0.25 μM). Cultures were stained 14 days later for neuronal marker MAP2 (*green*), aggregates of human α-syn (*red*) and cell nuclei (*blue*). Results are expressed as a percentage of the human α-syn staining area with PFF alone and show the mean ± SEM of 6 replicate cultures. A global analysis of the data was performed using one-way ANOVA followed by Dunnett’s multiple comparisons test. * *p* ≤ 0.05, ** *p* ≤ 0.01, **** *p* ≤ 0.0001 vs. PFF.

**Figure 11 biomedicines-10-02168-f011:**
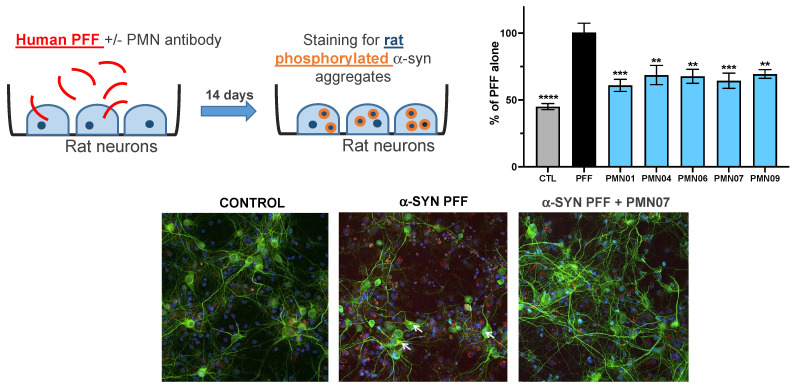
mAb inhibition of the recruitment of endogenous rat α-syn into phosphorylated aggregates. Cultures of primary rat hippocampal neurons were exposed to sonicated human PFF (1 μg/mL) without or with mAbs (0.05 μM, except for PMN09 at 0.25 μM). CTL = neurons incubated with vehicle alone. Cultures were stained 14 days later for neuronal marker MAP2 (*green*), aggregates of phosphorylated rat α-syn (*red*) and cell nuclei (*blue*). Results are expressed as a percentage of the phosphorylated rat α-syn staining area with PFF alone and show the mean ± SEM of 6 replicate cultures. A global analysis of the data was performed using one-way ANOVA followed by Dunnett’s multiple comparisons test. ** *p* ≤ 0.01, *** *p* ≤ 0.001, **** *p* ≤ 0.0001 vs. PFF.

## Data Availability

All of the relevant data (including Appendix A) are available in this publication.

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
