# Peer review of "Rational Generation of Monoclonal Antibodies Selective for Pathogenic Forms of Alpha-Synuclein"

_biomedicines, 2022, doi:10.3390/biomedicines10092168_

Round 1

Reviewer 1 Report

Rational generation of monoclonal antibodies selective for pathogenic forms of alpha-synuclein

The manuscript aims to explore the protective monoclonal antibodies (mAbs) against pathogenic forms of alpha synuclein. The authors implemented the computational modeling to identify the confirmational epitopes exposed on pathogenic alpha synuclein and selected corresponding mAbs. The findings in the present manuscript sound novel. The following are the concern about the present study.

1.               The authors showed the binding affinity of different mAbs to pathogenic species of α-syn. Is there any significant difference among the binding affinity of different mAbs (PMN3, PMN5, PMN7 and PMN8)?

2.               The authors should mention the statistical markers in Fig 5b and the respective legend.

3.               The authors are requested to check the manuscript for typo errors.

Author Response

Response to Reviewer 1 Comments

Point 1: The authors showed the binding affinity of different mAbs to pathogenic species of α-syn.  Is there any significant difference among the binding affinity of different mAbs (PMN3, PMN5, PMN7 and PMN8)?

Response 1: All antibody clones showed the same binding patterns and similar affinities for the different α-syn species.  This is now specified on line 328 of the manuscript.

Point 2: The authors should mention the statistical markers in Fig 5b and the respective legend.

Response 2: The Fig 5b legend now indicates that the error bars represent the standard error of the mean (SEM).

Point 3: The authors are requested to check the manuscript for typo errors.

Response 3: The manuscript was checked and any typographical errors were corrected.

Reviewer 2 Report

In the manuscript, entitled "Rational generation of monoclonal antibodies selective for pathogenic forms of alpha-synuclein” authors demonstrated that immunization with conformational epitopes of toxic -syn predicted by computational modeling allowed for the successful generation of highly selective mAbs capable of inhibiting the seeding activity and neurotoxicity of pathogenic -syn while avoiding detrimental interaction with normal monomers and physiologic tetramers. The manuscript as a whole is good but needs revision to increase its quality.

1.  In the whole manuscript, please identify the abbreviations when first mentioned.

2.      How the quantification of α-syn was calculated.

3.      What is the source of used antibodies, their catalog numbers and how was the antibodies' specificity validated? 

4.      The results need to be explained.

5.      The manuscript needs careful proof reading.

Author Response

Response to Reviewer 2 Comments

Point 1: In the whole manuscript, please identify the abbreviations when first mentioned.

Response 1: The manuscript was revised so that abbreviations are defined when first mentioned, and the abbreviation is used thereafter.

Point 2: How the quantification of α-syn was calculated.

Response 2: The molarity of α-syn in the various preparations was calculated based on the molecular weight of the monomer.  This information has been added to the Materials and Methods section (lines 178-180).

Point 3: What is the source of used antibodies, their catalog numbers and how was the antibodies' specificity validated?

Response 3: Catalog numbers have been added along with the commercial source of the antibodies.  Information on specificity is provided by the vendor along with data from publications that used the antibodies, in addition to our own experience with the antibodies (e.g., pan-reactive α-syn antibody 4D6 does show reactivity with all species of α-syn in Figures 4 and 6; the Abcam MAP2 antibody does stain cells with neuronal morphology in Figures 10 and 11, etc). 

Point 4: The results need to be explained.

Response 4: The results are explained at the end of each segment of the “Results” section.  For example, the first set of data ends with “Taken together, these results suggest that small misfolded regions (our predicted conformational epitopes) exposed on toxic α-Syn aggregates may directly contribute to their pathogenicity and represent biologically relevant targets for therapeutic antibodies”, and so on for the subsequent sets of data.  Results are also explained and placed into a greater context in the “Discussion” section.

Point 5: The manuscript needs careful proof reading.

Response 5: The manuscript was carefully proof-read and any typographical errors were corrected.

Reviewer 3 Report

This paper has developed the selective mAb against toxic-oligomer and soluble fibrils. The authors assessed the therapeutic efficacy of mAb using ThT assay, SPR assay, and immunofluorescent imaging. Although the authors provide a clear data for specificity of mAb for different forms of alpha-syn and anti-seeding activity of mAb, I would suggest authors to address several issues prior to its publication.

-          In experiment materials, the author should mention the length of recombinant monomeric human alpha-syn peptide obtained from rPeptide. Full-length or fragmented peptide?

-          In the condition to prepare the physiological tetramers, alpha-syn fibrils are also generated at same condition in vitro. To prepare the physiological tetramers, the authors should conduct SEC experiment using live cells containing alpha-syn protein.

-          PFFs (preformed fibrils) are made by tip-sonication of fibrils, and the length of PFF is important for cell-to-cell transmission, seeding and toxicity. Since the authors mention that the purchased PFFs are sonicated to generate small soluble fibrils, this reviewer wonders what size of PFFs they used. Excessive sonication of fibrils makes too small to elicit the Lewy-body pathology in vitro and in vivo. Please confirm the size of PFFs used in this study by TEM.

-          In spite of same condition (100 mM of alpha-syn), time-lapse ThT fluorescent units of monomer alone group are different in In Fig. 2, Fig. 5, and Fig. 8. The authors should be checking the condition or explain the inconsistence of results.

-          In Fig. 2b, Fig. 9 and Fig. 10, for comparison between group, the authors should perform the statistical analysis using multiple-comparison post-test rather than Student’s t-test.

-          In the Fig. 9, the authors present the survival of neurons as MAP2 staining after alpha-syn oligomers with or without mAbs. However, it would be better to use a cell death assay such as TUNEL for more accuracy of results.

-          In the Fig. 10 and Fig. 11, this reviewer thinks that 1 mg/ml of PFF is too high to use for cell culture. Many researchers have used PFF with a range microgram per mL (1-10 mg/ml) for cell experiments.

Minor comments,

1)      Please check any typos and grammar errors, there is a problem to use Greek symbols.

Author Response

Response to Reviewer 3 Comments

Point 1: In experiment materials, the author should mention the length of recombinant monomeric human alpha-syn peptide obtained from rPeptide. Full-length or fragmented peptide?

Response 1: The α-syn peptide was full length.  This information was added to the Materials and Methods section 2.2.

Point 2: In the condition to prepare the physiological tetramers, alpha-syn fibrils are also generated at same condition in vitro. To prepare the physiological tetramers, the authors should conduct SEC experiment using live cells containing alpha-syn protein.

Response 2: We appreciate this interesting suggestion from the reviewer.  However, for our purpose, we intentionally used synthetic, physiologic tetramers for a side-by-side comparison with synthetic toxic tetramers.  The size was confirmed by SEC and the physiologic tetramer preparations behaved in a manner consistent with that described for physiologic tetramers derived from cell lysates (no seeding activity, resistance to aggregation).  The use of purified protein has the additional advantage of avoiding potential participation of other proteins/aggregates with a similar molecular weight present in lysates which can complicate interpretation of the results.

In acknowledgement of this comment, we expanded on the text in the manuscript (lines 334-335): “In contrast, physiologic tetramer preparations behaved in a manner consistent with that described for physiologic tetramers derived from cell lysates and did not show seeding activity…”

Point 3: PFFs (preformed fibrils) are made by tip-sonication of fibrils, and the length of PFF is important for cell-to-cell transmission, seeding and toxicity. Since the authors mention that the purchased PFFs are sonicated to generate small soluble fibrils, this reviewer wonders what size of PFFs they used. Excessive sonication of fibrils makes too small to elicit the Lewy-body pathology in vitro and in vivo. Please confirm the size of PFFs used in this study by TEM.

Response 3: This is a good point from the reviewer which we clarified accordingly in the manuscript.  Details on the preparation of seeding soluble fibrils is now provided in Materials and Methods section 2.2, with references.  We used established sonication procedures to produce small soluble fibrils with seeding activity (as opposed to fully formed, full-size fibrils).  These sonicated preparations were confirmed to have seeding activity as demonstrated by our results in the ThT assay with synthetic protein (Figures 2a, 5, S3) and in cellular assays (Figures 10, 11), thereby achieving our goal.  

Point 4: In spite of same condition (100 mM of alpha-syn), timelapse ThT fluorescent units of monomer alone group are different in In Fig. 2, Fig. 5, and Fig. 8. The authors should be checking the condition or explain the inconsistence of results.

Response 4: This is an important point that was not made sufficiently clear in the original manuscript.  The difference is in the shaking protocol in the different studies.  In Figures 2 and 8, shaking was performed only for 30 sec prior to each hourly reading, while in Figure 5, continuous shaking was used to promote aggregation, giving rise to higher levels of aggregation with monomers alone.  This is described in Materials and Methods section 2.6 (“To be noted…” added to the text), and the legends of Figures 2, 5, and 8 state which shaking protocol was used.

Point 5: In Fig. 2b, Fig. 9 and Fig. 10, for comparison between group, the authors should perform the statistical analysis using multiple-comparison post-test rather than Student’s t-test

Response 5: As suggested by the reviewer, statistical analysis was revised using one-way analysis of variance (ANOVA) followed by Dunnett’s multiple comparisons test, as a more appropriate method. This is stated in Materials and Methods section 2.10.  Graphs and figure legends were also modified accordingly. 

Point 6: In the Fig. 9, the authors present the survival of neurons as MAP2 staining after alpha-syn oligomers with or without mAbs. However, it would be better to use a cell death assay such as TUNEL for more accuracy of results.

Response 6: Our goal was to quantitate surviving neuronal cells at the end of the culture period and look for differences with and without antibody. While TUNEL staining provides interesting information on cells dying from apoptosis at a given time point, it would not capture cells that already died prior to staining.  For our purpose, high-content imaging of MAP2-stained cultures was appropriate to provide statistically meaningful results on surviving cells.

Point 7: In the Fig. 10 and Fig. 11, this reviewer thinks that 1mg/ml of PFF is too high to use for cell culture. Many researchers have used PFF with a range microgram per mL (1-10 mg/ml) for cell experiments. 

Response 7: The reviewer is correct.  A concentration of 1 mg/ml would be excessive.  This is a case of the Greek symbol being lost along the way.  The concentration was 1 microgram per ml.  This has been corrected.

Point 8: Please check any typos and grammar errors, there is a problem to use Greek symbols.

Response 8: The manuscript was carefully proof-read and any errors were corrected.

Round 2

Reviewer 3 Report

The authors have addressed all of my comments. Great work.